# Potential of Drug Efficacy Evaluation in Lung and Kidney Cancer Models Using Organ-on-a-Chip Technology

**DOI:** 10.3390/mi12020215

**Published:** 2021-02-21

**Authors:** Seong-Hye Hwang, Sangchul Lee, Jee Yoon Park, Jessie Sungyun Jeon, Young-Jae Cho, Sejoong Kim

**Affiliations:** 1Department of Internal Medicine, Seoul National University Bundang Hospital, Seongnam 13620, Korea; jasno1@daum.net (S.-H.H.); lungdrcho@snubh.org (Y.-J.C.); 2Department of Urology, Seoul National University College of Medicine, Seoul 03080, Korea; slee@snubh.org; 3Department of Obstetrics and Gynecology, Seoul National University College of Medicine, Seoul 03080, Korea; jyparkmd08@snubh.org; 4Department of Mechanical Engineering, KAIST, Daejeon 34141, Korea; jsjeon@kaist.ac.kr; 5Department of Internal Medicine, Seoul National University College of Medicine, Seoul 03080, Korea

**Keywords:** organ-on-a-chip, microfluidics technologies, drug testing, cancer models, drug efficacy

## Abstract

Organ-on-a-chip (OoC) is an exponential technology with the potential to revolutionize disease, toxicology research, and drug discovery. Recent advances in OoC could be utilized for drug screening in disease models to evaluate the efficacy of new therapies and support new tools for the understanding of disease mechanisms. Rigorous validation of this technology is required to determine whether OoC models may represent human-relevant physiology and predict clinical outcomes in target disease models. Achievements in the OoC field could reveal exciting new avenues for drug development and discovery. This review attempts to highlight the benefits of OoC as per our understanding of the cellular and molecular pathways in lung and kidney cancer models, and discusses the challenges in evaluating drug efficacy.

## 1. Introduction

Animal models have been widely used to study the effects of drugs on specific diseases, but there are many differences between animals and humans. In the last decade, in vitro models have been improved with physiological systems that better reflect the mechanisms of specific human organs [1]. One of the main reasons for the high failure rates in clinical trials is the lack of toxicity and efficacy data, occurring due to the inability of existing preclinical models to recreate a biologically relevant human response. To build a reliable and practical drug-testing platform, it is critical to develop a model system that can reflect complex drug metabolism, real drug responses in a living system, and multiorgan interactions. All of these factors are considered to play a crucial role in determining the pharmacological effects of various drugs [2]. 

Recent advances in micro fabrication and tissue engineering have contributed to the improvement of organs-on-chips (OoC) that recapitulate the role of human organs. These “tissue chips” could be used for drug screening and safety testing to aid the drug development process in the early stages. They can also be utilized to model disease states, support new tools for the comprehension of disease mechanisms, and assess the efficacy of new therapies and pathologies [3]. 

## 2. Conventional Preclinical Methods

### 2.1. Two-Dimensional Cell Cultures 

During drug development, screening of the compound conduit is a key point in diminishing the number of molecules to a smaller pool and subsequently generating lead compounds; screening typically involves high-throughput methods where toxicity is evaluated in simple two-dimensional (2D) cellular models. For cancer research, in vitro 2D cultures have been regarded as the gold standard. However, recent research suggests that 2D cell cultures do not represent the actual in vivo conditions, and there is a clear shift to using three-dimensional (3D) cell cultures, with many extracellular matrices (ECMs) being suggested for this purpose. Despite the fact that these matrices, mostly derived from synthetic materials or extracted from animals, provide 3D structures, they still have fundamental limitations in the tumor microenvironment, including the absence of major factors present in humans [4]. 

### 2.2. Animal Studies

Cancer research using mouse models has gained popularity over the last few decades. Commercialization of these murine systems and sophisticated genetic manipulation technologies have made it possible to produce mouse models to study human diseases. In both the healthcare industry and academic research, there is a need for translatable cancer studies in animal models. Such a bench-to-bedside transition will provide an economically feasible and clinically effective long-term strategy for cancer treatment. However, the major limitation in utilizing mouse models as a translational platform is the lack of human cancer hallmarks, namely tumor genetic diversity and heterogeneity. Several patient-derived xenograft (also known as mouse avatars) models of different cancers and their therapeutic implications exist. Malaney et al. [5] reported two newly emerging concepts of coclinical trials and personalized mouse models known as “Mouse Avatars.” The development of mouse adjuvants involves the implantation of patient-derived tumor samples in mice for successive use in drug efficacy studies. Genetically engineered mouse models in coclinical trials are used in an ongoing human patient trial to guide therapy. Murine and patient trials are performed simultaneously, and information attained from the murine system is applied to the clinical management of the patient’s tumor. The trials allow for real-time integration of murine and human tumor data concurrently. In conjunction with diverse molecular profiling techniques, the coclinical trial and “Mouse Avatar” approaches have the potential to revolutionize healthcare processes and drug development. While these advanced murine models have been useful in understanding oncogenesis, a limitation related to the use of such inbred mouse models is the lack of heterogeneity found in human tumors. Intelligent use of conditional systems, chimeric mice, and inducible systems has partially offset this limitation; nevertheless, improvements regarding interpatient variability and tumor heterogeneity in drug responses are yet to be addressed [6].

## 3. Organ-on-a-Chip (OoC): Its Basic Elements and Comparison with Other Models

### 3.1. Basic Elements of OoC

#### 3.1.1. Cell Sourcing

The challenge of cell sourcing remains to be resolved in the OoC field. Currently, researchers can populate platforms with primary tissues from donors, commercially available immortalized cell lines, or induced pluripotent stem cells (iPSCs) (from either commercial sources or donors). Primary tissues from donors are optimal for OoC seeding, particularly for rare diseases. However, they may only be accessible in small amounts and are difficult to acquire from small population sizes or from diseased populations. When using cell lines with unknown donor demographics or primary cells from multiple individuals, increased genetic heterogeneity is seen in the resulting tissues, enhancing the variability of the results or producing confusing results [3].

The prospective source for most cells in OoC platforms are stem cells, specifically iPSCs. Technological progress has enabled the generation of renewable cell sources for various tissues. Tissue reprogramming using blood cells (to create iPSCs) or skin fibroblasts has many advantages over the usage of primary cells and provides remarkable opportunities. First, multiple tissue types, each with an isogenic background, can be generated from each individual. Second, both genetic therapy and tissue conversion aid the study of monogenic or Mendelian diseases by gene editing techniques, which were previously not possible [7]. Advances in iPSC hold promise for the verification of other population-wide studies. For instance, iPSC-derived fat and cardiac tissues were recently produced from entrants’ peripheral blood cells in the Framingham heart study cohort, and the genetic variant results coincided with those from previous studies. This is the first study to functionally confirm variants found in genome-wide association studies [8].

It is now widely known that the biomechanical environment used to culture stem cells can crucially affect their phenotype [9] and that 3D cultures encourage higher yields of differentiated stem cells and, in some cases, differentiation into more mature cell phenotypes [10,11,12]. One possible OoC application to advance the iPSC field was detailed recently [13]; here, the precisely restrained microfluidic environment of tissue chips enabled accurate exploration and control over the fluid environment surrounding differentiating cells. Such technological advances can lead to considerable progress in the stem cell field and could provide a solution for the OoC researchers’ need for renewable cell sources.

Differentiation protocols for iPS-derived cells largely differ between labs, and the resulting cells are similar to fetal phenotypes. Standardizing particular cell types would often be helpful; however, since this study field is comparatively young, it may take years before robust protocols are available. Epigenetic memory, in which iPS cells maintain certain characteristics of their primary phenotype, is also a factor that may influence phenotypic responses to potential therapeutics [14,15]. The pharmaceutical industry has started using neural iPSCs for lead optimization and validation, which, when used together with animal models and cell cultures, can yield meaningful results [16]. Patient-derived iPSCs in cancer research have been published, and tumor-derived iPSCs cross with other cancer models in diverse ways. iPSC-derived differentiated cells can be utilized to derive organoids, xenografts, immortalized cell lines, cocultures, and OoCs. Cocultures and OoCs may also integrate other iPSC-derived cell types [17].

#### 3.1.2. Blood Supply

Each tissue requires an appropriate supply of specific growth factors and nutrients; therefore, a key challenge is establishing a universal cell culture medium for linked OoC tissue systems or a “blood surrogate” [18]. For instance, circulating a 50:50 mixture of kidney-specific and liver-specific media in a connected liver-kidney system allowed the nephrotoxic metabolites of aristolochic acid to be evaluated [19]. Nevertheless, as the number of linked systems expands, the scaling solution success decreases, as every tissue ends up without an optimal culture medium, affecting the system’s function and physiological relevance. Approaches for linking systems may involve recirculating systems or creating a single-pass culture medium that can be supplemented or modified over time [20,21]. In addition, platforms engineered to enable tissue cultures in individual modules provide a path toward a circulating “blood mimetic” medium by including endothelial or synthetic barriers between the circulating medium and tissue modules [22,23,24]. Some scientists have approached the universal medium problem by supplying tissues with appropriate individual support mechanisms through surface chemistry alteration of the scaffold or platform on which cells are cultured when circulating in a general serum-free medium to institute fluidic flow to the system [25,26].

#### 3.1.3. Elements of the ECM

The ECM is a key element in a tumor’s cellular environment and a prevalent structural feature surrounding all eukaryotic cells. It also plays an essential role in tissue organization and cell signaling [27]. Besides understanding its composition, engineering a tissue also requires understanding the purpose of the scaffold or ECM and the functional cell interplay [28]. OoCs may utilize decellularized scaffolds or seed cells within synthetic or natural hydrogels to generate an environment suitable for cell growth. However, the 3D arrangement and ECM composition affect morphology, polarity, and cell survival [29,30,31]; therefore, they must be cautiously chosen and engineered to enable the formation of suitable tissue characteristics. The choice of ECM material must also be considered. Hydrogels are a widely used material owing to their biocompatibility, cell adhesion support, and resemblance to in vivo ECM and many soft tissues; however, they might be difficult to engineer and have insufficient standardized engineering protocols. Modeling complexities, even for comparatively simple tissues with few cell types, could be exponentially increased when innate or adaptive immune responses, vascularization, and frequent large variation in tissue sources between donors/suppliers/batches are involved. Recent advances in bioengineering have allowed new possibilities for biosensor integration into systems via ECM. For instance, incorporating fluorescent microgels, including peptides cleaved by specific enzymes [32], offers the opportunity to use ECM for real-time reading of OoC assay values. The ECM is a supporting network of macromolecules that provides biochemical and structural support to surrounding cells, while also generating biochemical cues for tissue growth and maintenance, and promoting cell adhesion and cell-cell communication. The ECM in animal tissues is composed of fibrous elements such as collagen and elastin, and tissue-specific and -linking proteins such as laminin and fibronectin [18].

#### 3.1.4. Scaling of Organs

A wide range of designs are intended for a wide range of platform sizes, from cell compartments resting on microscope slide-sized platforms of several hundred micrometers, to multiorgan systems of several centimeters. The most common systems are the recapitulation of downsized functional units of human organ systems, thousands or millions times smaller than the actual organ, and the adoption of microfluidic technology to provide fluid flow in accordance with the system to deliver nutrients and eliminate cellular waste, either by pump or gravity [3]. The main challenge regarding the platform integration is organ scaling between systems; in other words, organ systems need to manufacture in vivo outputs that are physiologically relevant to the other organs. For instance, if the liver module was five-fold “larger” than the kidney module, the linking system would not supply appropriate metabolite readouts. Allometric scaling can be utilized to quantify the relationship between diversely sized organs of diverse sizes. However, this type of “simple” scaling is impractical when considering the size differentials and complexities between human tissues and OoC [33]. Functional scaling may be a more appropriate strategy for determining meaningful ratios of organ masses, since tissue function is considered when designing the system (e.g., molecular filtration for kidney, gas exchange for lung). This more straightforward approach allows conservation of organ-specific functions at proper relative magnitudes, although relating platform results to in vitro−in vivo translation (or “in vivo physiology”) is still important to consider when physically linking platform systems [34].

### 3.2. Comparison with Other Advanced Cell Culture Techniques

#### 3.2.1. Conventional Preclinical Methods

Poor correlation between clinical and preclinical trial results due to ineffective preclinical models has resulted in the failure of many drug candidates to reach the market. Despite the fact that drugs are approved for clinical use, they have been later recalled because of severe kidney, liver, or cardiac toxicity [35]. Furthermore, inadequate side effect evaluation of some prescription drugs has caused the hospitalization of many patients [36]. Current preclinical research depending on animal models and 2D cell cultures must be improved to diminish drug development costs and advance patient outcomes [37].

Tissue chips are devices designed to arrange cells in a 3D structure that imitates the organ’s functions and responds physiologically upon exposure to drugs, cell signaling molecules, hormones, and biomechanical stressors. Platforms vary in design, grow in a structurally defined manner, and have cellular arrangements of numerous cell types. However, it is difficult to apply these design features in two dimensions. Moreover, chips are designed from materials that enable cells to be microscopically visible, allowing for longer real-time monitoring and imaging of cell function and health. This longitudinal study method allows the modeling of the recovery and time course of drug responses as well as the effects of periodic hormonal exposures over time. Furthermore, the system’s fluid flow allows the outflow to be collected for biochemical or enzymatic assays. The diversity of these platform designs can be utilized in novel and innovative ways to address a wide range of biological questions [3].

#### 3.2.2. Comparison with Organoid Models

An organoid is a three-dimensional construct consisting of multiple cell types and it is able to simulate the functionality and architecture of native organs. Organoids are effective for both in vivo and in vitro studies and represent one of the most recent model innovations to epitomize the physiological processes of whole organisms [38]. Organoids have many advantages compared to traditional two-dimensional cultures. They can display near physiological behaviors and cellular composition. Many organoids can pass through extensive cell culture expansions and sustain genome stability [39,40,41,42], making them adequate for biobanking and high-throughput screening [43]. Compared to animal models, organoids decrease experimental complexity and allow the study of disease aspects and human development that are not easily or precisely modeled in animals. As a bridge between animal models and conventional 2D cultures, organoids have multilateral advantages that capture biological complexity and experimental manipulability [44,45,46,47].

Organoids can be generated from somatic cells, pluripotent stem cells, or adult stem cells, including progenitor cells [46]. As organoid technology expands the biomedical research potential, there is an emerging need for progressive engineering approaches for analyzing organoids and their microenvironment, production, and control. OoC technology has the potential to fulfill this need, as it can be used to investigate how organoids may be affected and address the main technical challenges in organoid research. Emerging opportunities and future limitations for the application and development of OoC technology are discussed. To overcome the limitations of stem cell research, researchers using conventional culture techniques and developmental biology are teaming up with physicists and engineers to develop unique in vitro technologies for organoid research. At the foremost of this undertaking is organoid consolidation using OoC technology [48].

#### 3.2.3. Comparison with 3D Bioprinting

Bioprinting offers the ability to create highly complicated 3D structures using living cells. This cutting-edge technique has gained considerable popularity and applicability in various fields. Bioprinting methods have been improved to rapidly and effectively pattern living cells, biomaterials, and biological macromolecules. These technologies hold great potential for applications in cancer research. Bioprinted cancer models show a significant advancement over previous 2D models by imitating 3D complexity and allowing the modeling of physiologically relevant cell-matrix and cell-cell interactions. Bioprinting methods are based on inkjet, laser technologies, and micro extrusion. Bioprinted models that simulate the tumor microenvironment offer a platform for better understanding of cancer treatment, cancer pathology, and anticancer drug screening [49].

Most 3D printing techniques are unable to print truly “freeform” items without spatial limitations on the object’s shape; this is due to the inability to deposit material in a region with no direct connection to a previous item section. Extrusion and inkjet bioprinting share numerous design limitations pertaining to 3D printing. Key limitations of these novel tumor-engineering applications include optimization of fluid mechanisms for material extrusion and material phase alteration after extrusion. Three-dimensional hydrogel support bath and template casting are excellent techniques for constructing clear in vitro 3D vessel structures, but with limitations. The bulk hydrogel cast surrounding the sacrificial material will be equal to the ECM material and cellular composition and therefore will not recapitulate spatially heterogeneous native tissue. Existing techniques can only utilize a few biomaterials and current 3D printing technologies are able to manufacture vessel diameters on the base of 100 μm and thus cannot acquire capillary level resolution within 10 μm. There are still limitations to the use of 3D printing for investigating metastasis. Therefore, the development and optimization of biomaterials to improve their properties is needed [50].

OoC engineering aims to generate artificial living organs that simulate the physiological and complex responses of real organs to investigate drugs by precisely studying the cells and their microenvironments. To accomplish this, artificial organs should be created with various cell and ECM types and should recapitulate functions, cell differentiation, and morphogenesis according to the native organ. An advantageous strategy is 3D printing, which regulates the layer-by-layer cell assembly and spatial distribution, ECMs, and other biomaterials. Owing to this unique feature, the incorporation of 3D printing into OoC engineering will facilitate the modeling of tissue-specific functions, micro-organs with heterogeneity. Furthermore, fully 3D-printed OoC more easily facilitate other mechanical and electrical elements with chips and can be implemented via automated massive production. The potential of 3D cell-printing technology and the recent advances in engineering OoC suggest the prospects of these technologies to create highly useful and reliable drug-screening platforms. Many leading companies work on the 3D printing of chip devices or artificial tissues, but the 3D-printed OoC remains to be commercialized. Progression in printing technologies will accelerate the practical use of these techniques in drug development and the commercialization of tissue/organ models to overcome several refractory diseases in humans [51].

## 4. Potential of OoC as a Drug Efficacy Evaluation in Lung and Kidney Cancer Models

### 4.1. OoC as Advanced Microfluidic Technologies of the Lung and Kidney Model

Researchers have assessed metabolism-dependent drug toxicity and efficacy on a multilayer OoC. This in vitro model OoC represents different tissues simultaneously and allows the characterization of the dynamic metabolism of anticancer drugs. It also provides a simple method for evaluating drug bioactivity in various target tissues, suggesting its usefulness in pharmacodynamic/pharmacokinetic studies, drug toxicity and efficacy testing [52]. Caballero et al. [53] described how the tumor-vessel-on-a-chip technique can be applied to study targeted drug delivery and the main factors required for the design of these materials. Its role in driving forward the next generation and future applications of this approach of targeted drug delivery systems will be discussed.

OoC technology can simulate the physiological and pathological microenvironments of organs and tissues in vitro, thereby eliminating the use of animal models for predicting drug efficacy and toxicity. Yang et al. [54] developed a lung-on-a-chip in 2018. The chip is simple, effective, and easy to utilize, therefore it is expected to play a role in tissue engineering and clinical treatments and have important applications for the personalized treatment of lung tumors. The merging of multi-organ-on-a-chip (MOC) technology with 3D in vitro models has taken in vitro chemical evaluations to an exceptional level. By connecting multiple organotypic models, MOC allows for the crosstalk between different organs to be studied to estimate a compound’s efficacy and safety better than in single cultures. The lung/liver-on-a-chip platforms provide new opportunities to identify efficacy and safety or to investigate the inhaled aerosol toxicity of new drugs targeting the human lung [55].

Kim et al. [56] reported a pharmacokinetic profile that reduces nephrotoxicity of gentamicin in a perfused kidney-on-a-chip platform (Figure 1). Lee and Kim [57] reviewed studies that involved the generation of experimental environments similar to the physiological environments in human organs using kidney-on-a-chip models and obtained experimental results that better manifest human physiology. Kidney-on-a-chip models can be used to overcome the drawbacks of traditional animal models and to more effectively identify drug efficacy, drug-induced nephrotoxicity, and interactions.

### 4.2. OoC as a Disease Model for Drug Efficacy Evaluation

Once toxicity screens are complete, drugs are tested in animals and humans, but due to insufficient efficacy data, some drugs do not produce the expected clinical effects [58]. Human volunteers may suffer potentially catastrophic and unexpected side effects, assuming efficacy data is not properly demonstrated before the first in-human trials [59]. Between 2005 and 2010, Cook et al. examined why some drug development programs at AstraZeneca failed and found that insufficient efficacy data led to failure rates of 57%–88% in phase II clinical trials [60].

OoC platforms can help decrease the failure rate in numerous ways. For instance, they can help screen out detrimental compounds earlier, so fewer but more promising compounds can reach human trials. In addition, tissue platforms represent a variety of human organ systems that can be used to screen for drug efficacy before reaching clinical trials, and mechanisms of action can be identified and modeled appropriately. Ethics is another advantage of using OoC in early drug development, as it could considerably reduce animal use in preclinical stages. Animal models are still needed because they represent whole organisms, but as integrated and individual organ microsystems become more easily available, cheaper, and suitable for medium- to high-throughput screening, animal testing rates in drug development may significantly decrease. By incorporating an oxygen sensor, a microfluidic platform was used together with physiology-based pharmacokinetic (PBPK) modeling to forecast the absorption, distribution, metabolism, and excretion (ADME) of chemicals. This early system was useful for predicting effective drug dosages and concentrations in human and animal studies. There will also be continued progress of “body-on-a-chip” systems for multiorgan diseases; therefore, there is great potential for this exciting and rapidly expanding field to provide insight for the pharmaceutical industry and biomedical researchers regarding treatments for the most intractable and prevalent disorders of our time [3].

In addition to helping to understand toxicity in human tissues, OoC also allows the modeling of disease states, thereby allowing mechanistic observation of not only drug efficacy but also disease pathology and potential therapeutic off-target effects. The potentially enhanced comprehension of human disease physiology from disease models on OoCs could help resolve the high attrition rates of prospective compounds in lead optimization and clinical stages due to lack of efficacy [60].

### 4.3. Elements of the Drug Efficacy Evaluation in Lung and Kidney Cancer

#### 4.3.1. Assays for Drug Efficacy Evaluation in OoC

Conventionally, the drug efficacy assessment for 3D cells is fulfilled by staining techniques such as cell counting kit-8 (CCK-8), 3-(4, 5-dimethylthiazol-2-yl)-2, 5-diphenyl tetrazolium bromide (MTT), or water-soluble tetrazolium salts-1 (WST-1) assay. These trials only indicate whether the cells survived, but they are unable to monitor the cells during real-time long-term recordings. In addition, once the cells are stained, they are not able to be used for further treatments. The cell electrical impedance reflects the characteristics, such as cell number, cell morphology, and cell attachment on electrodes in a continuous and dynamic way [61,62], which has a wide range of applications, such as drug efficacy and toxicity assessment [63,64]. Wu et al. [65] reported on the bionic 3D spheroids biosensor chips for high-throughput and dynamic drug screening and assessed cell viability with fluorescence microscopy using the Calcein-AM/propidium iodide (PI) Double Stain Kit (Yeasen), where live cells were stained green by Calcein, and dead cells were stained red by PI. Ki-67 immunostaining [66], cell counting [67] and 5-ethynyl-2-deoxyuridine (EdU) labeling [68] are used for the proliferation assay. Gene expression can be evaluated by real-time quantitative PCR [67] in OoC. Esch et al. [69] reported about OoC at the frontiers of drug screening and reviewed many other assay-related complex biological processes in this technology.

#### 4.3.2. Epidermal Growth Factor Receptor (EGFR) and Vascular Endothelial Growth Factor (Vegf) Related Molecules as Key Biomarkers of Lung and Kidney Cancer

The human epidermal growth factor receptor (EGFR) is a main regulator of organ homeostasis, mediating adult cell proliferation and differentiation [70]. EGFR-targeted antibody therapies are being increasingly applied in cancer therapy since EGFR-tyrosine kinase activation is the main pathway mediating lung cancer progression [71,72]. EGFR-targeted therapies such as the inhibition of specific tyrosine kinases, monoclonal antibody use, or a combination of both are promising approaches [73,74]. Anti-EGFR antibodies are known to influence EGFR-mediated signaling pathways, such as soluble EGFR and its ligands EGF and tumor growth factor (TGF)-alpha by inducing apoptosis and growth arrest, decreasing cell migration, increasing cell differentiation and attachment [75].

Renal cell carcinoma (RCC) is responsible for up to 85% of kidney cancers, and over the past 12 years, RCC treatment has transitioned from a nonspecific immune treatment to a novel immunotherapy, namely vascular endothelial growth factor (VEGF) targeted therapy [76,77]. The inactivation of the von Hippel-Lindau (VHL) tumor suppressor gene leads to increased activity of the hypoxia-induced factor (HIF) in the tumor cell and eventually to the overexpression of platelet-derived growth factor and VEGF [78,79]. HIF activity may also be enhanced by targeting of the mammalian rapamycin (mTOR) pathway [80].

### 4.4. Comparison with Conventional Efficacy Testing Methods and Institutional Devices

#### 4.4.1. Comparison with Animal Testing

The major assumption justifying animal testing is that animal models help predict human responses in toxicology research and drug discovery [81]. Some animal study failures can be due to a lack of standardized experimental conditions; however, the majority occur due to the invalidity of this assumption [82]. Animal models have the advantage of studying system-wide drug effects and multiorgan interactions. However, the inbreeding of numerous laboratory animals restricts the genetic variability, and these models are not representative of human population diversity. Laboratory animals are genetically similar; therefore, it is often difficult to draw population-level conclusions on drug efficacy and safety or disease mechanisms from these studies. In contrast, iPSCs can be derived from various patient populations, creating data that are more representatively related to the human population [83].

#### 4.4.2. Comparison with Clinical Trials

Technological advances with induced pluripotent stem cells and OoC have the potential to overcome the challenges of drug development. OoC may do so by offering methods for performing “clinical trials-on-chips” (CToCs) to observe the design and execution of rare disease clinical studies, which otherwise would be impossible with other culture systems. If applied properly, CToCs can considerably impact clinical trial design with regard to anticipated key outcomes, risk and assessment of clinical benefit, safety and tolerability profiles, value and efficiency, population stratification, and scalability. The OoC models may play a pivotal role in streamlining the clinical trial process. The integration of stem cell engineering and OoC technology can improve the development of personalized models and predict patient-specific toxicity and efficacy. This could lead to more effective clinical trials with remarkably reduced preclinical testing requirements. Such personalized models may also be useful in individualized dosing regimens based on patient-specific pharmacokinetics and exploring patient-specific biomarkers [84].

#### 4.4.3. Institutional Devices as Alternatives to Animal Testing

The European Centre for the Validation of Alternative Methods advocated a formal validation study on in vitro methods to predict skin corrosivity in 1996 and 1997. The concurrences between the skin corrosivity classifications originating from the in vitro data were high, and the test was able to discriminate between corrosive and noncorrosive chemicals for every chemical type studied [85]. The development and validation of systemic alternatives to animal testing is critical not only from an ethical perspective, but also to improve safety decision-making with mechanistic information with higher relevance to humans. In 2009, the International Cooperation on Alternative Test Methods was founded by validation centers from Europe, the USA, Canada, and Japan. Korea joined in 2011 and, together with Brazil and China, currently acts as an observer [86].

A multilaboratory validation study of the vitrigel-Eye Irritancy Test (EIT) method was conducted to assess EIT as an alternative to in vivo eye irritation testing. After a thorough data review for the method to be used for regulatory purposes, it was demonstrated that a more defined applicability domain improved the false negative rate. Within this prudently defined applicability domain, these results suggested that the vitrigel-EIT method could be a valuable alternative for differentiating optic nonirritant test chemicals from irritants [87].

The National Center for Advanced Translational Science (NCATS) has been developing tissue-on-a-chip for drug screening programs since 2011, in collaboration with the National Institutes of Health, the Defense Development, and the US Food and Drug Administration [88]. The NCATS announced a plan to develop organ chips for efficacy testing and disease modeling that could support further discovery of human disease tissue chip models emulating the pathology of major human organs and tissues in October 2016 [89]. Recently, Lee and Lee [90] analyzed the scientific and technological trends based on cutting-edge models and reported that the biomimetic tissue chips are developed for utility as nonanimal models for testing efficacy and safety at the nonclinical stages of drug discovery.

### 4.5. Potential of Drug Efficacy Testing in Tumor-on-a-Chip and Metastasis-on-a-Chip

Shirure et al. [91] reported that a tumor-on-a-chip platform to investigate drug sensitivity and progression could provide avenues to improve our understanding of tumor metastasis and a microfluidic platform that imitates biological mass transport close to the arterial end of a capillary was included. A central feature was a dormant, perfused 3D microvascular network generated before loading the patient-derived tumor organoids or tumor cells in an adjacent compartment. Physiological delivery of drugs and/or nutrients to the tumor occurs through the vascular network. Finally, by evaluating the changes to chemo- and antiangiogenic therapies, the study group reported the platform’s potential to be used for personalized medicine and drug discovery. Precision medicine-based cancer treatments can only be discovered if individual tumors are rapidly evaluated for therapeutic sensitivity in a clinically appropriate timeframe (≤14 days). The platform indicated that this is possible and provided convincing information for the advancement of cancer precision medicine. Recently, Chramiec et al. [92] reported the integrated human organ-on-a-chip model for predictive studies of antitumor drug efficacy and cardiac safety.

Metastasis is one of the most critical factors leading to poor cancer prognosis, and effective suppression of primary cancer cell proliferation in a metastatic site is the most efficacious method for preventing cancer progression. However, there is a shortage of biomimetic 3D in vitro models that can closely mimic the continual growth of metastatic tumor cells in an organ-specific ECM for evaluating effective therapeutic strategies. According to recent research, Wang et al. [93] reported a novel 3D metastasis-on-a-chip model that imitates the progression of kidney cancer cells transferred to the liver to predict treatment efficacy. This article demonstrated that a tumor progression model based on metastasis-on-a-chip with organ-specific ECM would be a valuable tool for rapidly developing new chemotherapeutic agents to estimate treatment regimens. Furthermore, the tumor progression model can be utilized to optimize dosage regimens, assess anticancer efficacy, and establish 3D metastatic cancer models. Xu et al. [94] reported the acquired drug resistance in lung cancer derived brain metastasis based on a multiorgan microfluidic model and Oliver at al. [95] reported quantifying the brain metastasis tumor micro-environment using an organ-on-a-chip 3D model, machine learning, and confocal tomography.

Lee et al. [96] reported a 3D microfluidic platform to recapitulate angiogenic sprouting when coculture with diverse cancer cell types and tumor vascular mapping for evaluating antiangiogenic nanomedicine in 2021. This model enables efficient and rapid evaluation of antiangiogenic nanomedicine and provide a powerful platform for the discovery of effective and safe nanomedicine for cancer therapy.

## 5. Conclusions

OoC are not common solutions, and other methods will still be preferred for modeling certain in vivo processes. Despite the fact that OoC offer advanced biological modeling, there are major limitations that they may never circumvent; thus, alternative tools may be preferred. Despite their limitations, OoC have the potential to transform drug development and discovery. Rigorous validation of this technology with not only animal data but also clinical trial results is required to determine whether OoC models have predictive capability and represent human-relevant physiology across clinical outcomes and various drug classes. Achievements in the OoC field could reveal exciting new avenues for drug development and discovery. Much remains to be done; thus, there are still many opportunities to discover the tremendous possibilities that OoC technology has to offer.

## Figures and Tables

**Figure 1 micromachines-12-00215-f001:**
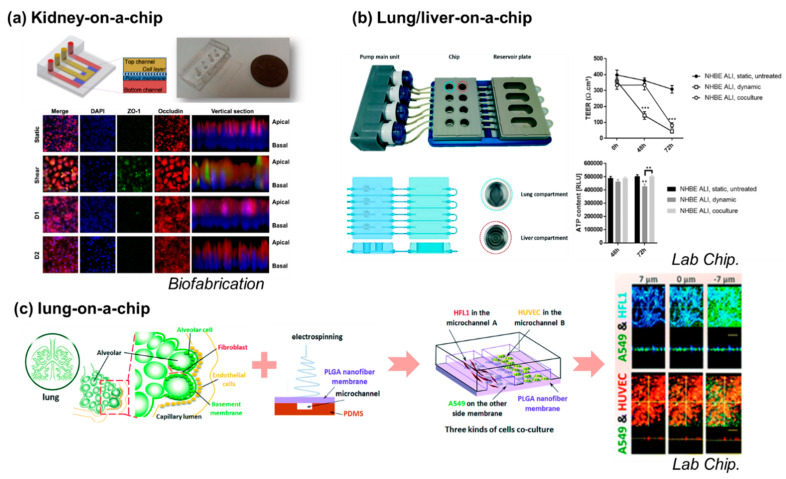
Organ-on-a-chip models for mimicking a single organ or a combination of organs. (**a**) Kidney-on-a-chip is developed for monitoring nephrotoxicity. Schematic design and actual image of a kidney-on-a-chip. Junctional protein expression of each group. The static and shear groups are measured before exposure to gentamicin, and D1 and D2 groups are measured 24 h after exposure to gentamicin. All groups show improved polarization compared to Transwell cultures. Source: *Biofabrication,* 2016 [56]; (**b**) lung/liver-on-a-chip combines two types of organs for investigating aerosol toxicity. A photograph of the chip system comprising the pump main unit with four pump heads, the PEEK chip, and the reservoir plate. A schematic view of the chip comprising four circuits is shown; each circuit includes two compartments to house the lung and liver tissues. The cross-section schemas of the plates show the path of the tubes and channels and the relative depth of each well. Source: *Lab Chip,* 2018 [55]; (**c**) a close-up view of the two compartments showing the groove pattern on the bottom of the wells; left (**c**). AFB1 toxicity was assessed using the lung/liver-on-a-chip. TEER and ATP content were measured in untreated or AFB1-exposed NHBE ALI tissues either in monoculture (NHBE ALI, dynamic) or coculture with HepaRG spheroids (NHBE ALI, coculture); right (**c**) nanofiber-membrane-assisted lung-on-a-chip system is used for anticancer drug testing. Source: *Lab Chip,* 2018 [54].

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
