# Peer review of "Potential of Drug Efficacy Evaluation in Lung and Kidney Cancer Models Using Organ-on-a-Chip Technology"

_micromachines, 2021, doi:10.3390/mi12020215_

Round 1
Reviewer 1 Report
The authors wrote a comprehensive review manuscript detailing how organ-on-chips can be used as models to evaluate drug efficacy in pre-clinical trials. The manuscript is well written and provides an interesting outlook to the potential applications of organ-on-chips, while comparing OoCs to other existing models and technologies.
However, the manuscript reads more like a book chapter than like a review paper. There is, in my opinion, a lack of adequate references related to the work that exists currently in the literature. Regarding the scope of this Journal (Micromachines), I would expect to see more examples of existing organ-on-chip models, how they differ technically, and which pros and cons they offer between themselves and when compared to other technologies. I also consider the title misleading as the manuscript is very general and does not focus solely on lung and kidney cancer-on-chip models, as it would make one believe. Thus, while I consider this manuscript adds a contribution to the field, I believe that it needs considerable re-structuring and the addition of relevant references of existing models of OoCs together with their discussion.
Other more specific comments include:
line 17: "Many opportunities that OoC technology has to offer remain." This sentence on its own is disconnected and does not make sense.
line 22: "efficacy" doesn't seem like a fitting keyword
line 26: There many other differences between animals and humans beyond gene and protein modulation.
line 27: 'recapitulate' would be a better word than 'synopsize' here
Section 3.1 starts immediately by giving examples of a couple OoC applications, including a 'multilayer OoC'. However, at this point in the manuscript it still hasn't been detailed what constitutes an OoC. In fact, I believe this is not done throughout the whole manuscript and it definitely should be done before section 3.1.
line 86: Which 'in vitro model'? This sentence is not properly introduced.
line 96: Again, which 'developed lung-on-a-chip' is being referred to here? The sentence needs to be rephrased.
line 107: "Kidney-on-a-chip" models...
line 166: two types OF organs ... there is also an extra space between 'toxicity' and 'A'
line 127: Cells
line 150: in comparison to what?
line 175: What does 'scaling mixtures of endothelial barriers' mean?
line 192: eukaryotic cells
line 235: double full stop at the end of the sentence
line 260: organoids don't arise necessarily from stem cells
line 269: 'are suitable for real-time imaging techniques'. Depends on the size of the organoid and where they are cultured, correct?
line 309: What does 'special properties' mean?
312: Biomaterials are not a drawback
line 323: What does 'circular movement within a microfluidic device mean'?
lines 351-353: Which 'early platform' is being referred to here?
line 368: The examples given are not so much to determine live/dead cells but to quantify cell metabolic activity, which is not quite the same thing.
line 373: "Therefore, cell viability was assessed with fluorescence 373 microscopy using the Calcein-AM/propidium iodide (PI) Double Stain Kit (Yeasen)," I don't understand where this comes from. It seems like a section from Materials&Methods of an original research paper.
line 377: "were used". Where were they used?
The construction of section 4.2.1 seems all wrong to me.
line 384: All of these methods can also be analysed in platforms other than organ-chips. What would be interesting would be to showcase how these analyses are typically interfaced with OoCs and to provide examples from the literature.
Section 4.2.3 and Table 2 seem completely disconnected to organ-on-chips.
Section 3.1 seems disconnected to the rest of the paper. I suggest moving 3.1
line 438: wrong paragraph insertion.
line 479: Which lung tumour-on-chip? There's a 16-line paragraph mentioning a lung tumour-on-chip with a single reference to another review paper at the end. As a review, this manuscript should with a clear reference to the lung tumour-on-chip paper together with a thorough discussion, especially because kidney and lung tumour-on-chips make the title of this manuscript.
Author Response
Response to Reviewer 1 Comments
The authors wrote a comprehensive review manuscript detailing how organ-on-chips can be used as models to evaluate drug efficacy in pre-clinical trials. The manuscript is well written and provides an interesting outlook to the potential applications of organ-on-chips, while comparing OoCs to other existing models and technologies.
However, the manuscript reads more like a book chapter than like a review paper. There is, in my opinion, a lack of adequate references related to the work that exists currently in the literature. Regarding the scope of this Journal (Micromachines), I would expect to see more examples of existing organ-on-chip models, how they differ technically, and which pros and cons they offer between themselves and when compared to other technologies. I also consider the title misleading as the manuscript is very general and does not focus solely on lung and kidney cancer-on-chip models, as it would make one believe. Thus, while I consider this manuscript adds a contribution to the field, I believe that it needs considerable re-structuring and the addition of relevant references of existing models of OoCs together with their discussion.
We are grateful for this valuable comment and appreciate the positive feedback.
Several statements that we made were more ambiguous than intended and we have adjusted the text to be clearer. We realize that the initial text may have been unclear and acknowledge that more detail is needed, we have added references specifically and revised order and structure of section 4.
Several references related to the efficacy that exists currently in organ-on-chip model have described in the last part of section 4 and we have tried to focus on the potential of OoC as efficacy evaluation in lung and kidney cancer-on-chip models.
Other more specific comments include:
line 17: "Many opportunities that OoC technology has to offer remain." This sentence on its own is disconnected and does not make sense.
This is an excellent point, we decided to delete the ambiguous sentence.
line 22: "efficacy" doesn't seem like a fitting keyword
We agree and have added recently published references related to efficacy evaluation in OoC, especially focused on Lung and kidney cancer models to the last part of section 4.
line 26: There many other differences between animals and humans beyond gene and protein modulation.
We agree that it was only part of the content.
We deleted “in terms of gene and protein modulation.” and changed the sentence simply.
line 27: 'recapitulate' would be a better word than 'synopsize' here
We changed the better word 'recapitulate'. : line 35 on page 1 in revised manuscript
Section 3.1 starts immediately by giving examples of a couple OoC applications, including a 'multilayer OoC'. However, at this point in the manuscript it still hasn't been detailed what constitutes an OoC. In fact, I believe this is not done throughout the whole manuscript and it definitely should be done before section 3.1.
This suggestion is valid, so we moved and changed the section from 3.1. to 4.1
line 86: Which 'in vitro model'? This sentence is not properly introduced.
'in vitro model' means Ooc in previous sentence, so I added comma and Ooc after 'in vitro model' line 82on page 2 in revised manuscript
line 96: Again, which 'developed lung-on-a-chip' is being referred to here? The sentence needs to be rephrased.
We revised the sentence more sepcifically and added the refernce number and year of publish as follows. Yang et al. [54] developed a Lung-on-a-chip in 2018
line 299 on page 7 in revised manuscript
line 107: "Kidney-on-a-chip" models...
We added “models : line 313 on page 7 in revised manuscript
line 166: two types OF organs ... there is also an extra space between 'toxicity' and 'A'
We deleted an extra space and put a period at the end of a sentence.
line 326 on page 7 in revised manuscript
line 127: Cells
We added “sourcing” to the subtitle only “Cell” : line 82 on page 2 in revised manuscript
line 150: in comparison to what?
It is considered an unnecessary expression of the text, so we decided to delete.line 105 on page 3 in revised manuscript
line 175: What does 'scaling mixtures of endothelial barriers' mean?
We intended to describe the meaning that blood supply issue in OoC means circulating media mixture in case of Multi-Organs-on-Chips. The expression is an ambiguous in the sentence, we delete : line 129 on page 3 in revised manuscript
line 192: eukaryotic cells
We changed the word, from eukaryotes to eukaryotic cells.
line 146 on page 4 in revised manuscript
line 235: double full stop at the end of the sentence
We deleted one the other full stop. : line 189 on page 4 in revised manuscript
line 260: organoids don't arise necessarily from stem cells.
According to advice, I deleted the sentence. line 213 on page 5 in revised manuscript
line 269: 'are suitable for real-time imaging techniques'. Depends on the size of the organoid and where they are cultured, correct?
Because it’s inaccurate, I deleted the sentence : line 221 on page 5 in revised manuscript
line 309: What does 'special properties' mean ?
We deleted the words because it was not considered essential expression in the flow.
line 262 on page 6 in revised manuscript
312: Biomaterials are not a drawback
We deleted the sentence. : line 265 on page 6 in revised manuscript
line 323: What does 'circular movement within a microfluidic device mean'?
The circular movement means flow of media and a microfluidic device means OoC system.
We deleted an ambiguous expression : line 276 on page 6 in revised manuscript
lines 351-353: Which 'early platform' is being referred to here?
The reference published in 2017, we intended to express for 'early platform' means developed platform before the year. However, we considered the “early” is not essential word, so I revised. : line 359 on page 8 in revised manuscript
line 368: The examples given are not so much to determine live/dead cells but to quantify cell metabolic activity, which is not quite the same thing.
We revised the structure and subtitle of section 4.2 and we decided to delete the expression ” live/dead”
line 373: "Therefore, cell viability was assessed with fluorescence 373 microscopy using the Calcein-AM/propidium iodide (PI) Double Stain Kit (Yeasen)," I don't understand where this comes from. It seems like a section from Materials & Methods of an original research paper.
We revised that Wu et al. [64] reported the bionic 3D spheroids biosensor chips for high-throughput and dynamic drug screening and assessed cell viability with fluorescence microscopy using the Calcein-AM/propidium iodide (PI) Double Stain Kit (Yeasen), where live cells were stained green by Calcein, and dead cells were stained red by PI.
line 377: "were used". Where were they used?
We added the specific refernces and revised. : line 387 on page 9 in revised manuscript
Ki-67 immunostaining [66], cell counting [67] and 5-ethynyl-2-deoxyuridine (EdU) labeling [68] were used for the proliferation assay.
line 387 on page 9 in revised manuscript
The construction of section 4.2.1 seems all wrong to me.
We revised the subtitle and structure of section 4.2
4.2 OoC as a disease model for drug efficacy evaluation
Section 4.2.1 and 4.2.2 and integrated and changed the subtitle to 4.3.1
4.3. Elements of the drug efficacy evaluation in lung and kidney cancer
4.3.1 Assays for the drug efficacy evaluation in OoC.
line 384: All of these methods can also be analysed in platforms other than organ-chips. What would be interesting would be to showcase how these analyses are typically interfaced with OoCs and to provide examples from the literature.
We added the specific references related to the assays in OoC platform.
Gene expression can be evaluated by real-time quantitative PCR [67] in OoC. Esch et al. [69] reported about OoC at the frontiers of drug screening and reviewed many other assays related complex biological processes in this technology.
line 388 on page 9 in revised manuscript
Section 4.2.3 and Table 2 seem completely disconnected to organ-on-chips.
This is a good point, we decided to delete table 2 and changed the subtitle of section 4.3.2.
EGFR and VEGF related molecules as key biomarker of lung and kidney cancer
line 392 on page 9 in revised manuscript
Section 3.1 seems disconnected to the rest of the paper. I suggest moving 3.1
This is an excellent point. We moved the section 3.1 to section 4.1
line 438: wrong paragraph insertion.
We checked and deleted the wrong paragraph insertion.
line 429 on page 9 in revised manuscript
line 479: Which lung tumour-on-chip? There's a 16-line paragraph mentioning a lung tumour-on-chip with a single reference to another review paper at the end. As a review, this manuscript should with a clear reference to the lung tumour-on-chip paper together with a thorough discussion, especially because kidney and lung tumour-on-chips make the title of this manuscript.
The paragraph is more ambiguous than intended. We agree that the paragraph contained too much content and references and consider that organ-tumor-on-a-chip is not essential. According to the title of section 4.5 Potential of drug efficacy testing in tumor-on-a-chip and metastasis-on-a-chip, we deleted the paragraph and focused on tumor-on-a-chip and metastasis-on-a-chip.
Instead of the the paragraph, we added recently published references related to efficacy evaluation in OoC (line 479 page on 10 in revised manuscript) and especially focused on Lung and kidney cancer models to the last part of section 4.

Reviewer 2 Report
Comment on manuscript micromachines-1099538
Title: Drug Efficacy Evaluation in Lung and Kidney Cancer Models Using Organ-On-A-Chip Technology
This manuscript reviews the recent progress of organ-on-chip technology, compares its application/performance with conventional 2D and 3D culture on drug development usage, and also discuss the limitation of the technology.
The manuscript is well organized and the description is mostly clear and concise.
The reviewer has some suggestion on minor problems.
The font of the terms in-vivo and in-vitro needs to be consistent. In some places, they are in the plain font but in other places, they are italicized. Whether italicized or not needs to be consistent.
In line 27, the meaning of the sentence is not clear: “…data acquired in animal studies is not identical to the mechanisms in humans.”
In line 66, “…heir therapeutic implications exist, and Malaney et al. reported…”, it is not clear whether a reference is to be inserted.
Figure 1 needs some modification in the numbering. There are more than one (c)’s in the figure. This causes confusion in the description in the caption. This issue must be taken care of.
Line 163, it is not clear what do the authors mean by “…fast nature of iPSC…”. A more elaborative description is needed.
In lines 274 to 277, it is stated that “… they are less frequently used than stem cell organoids. However, the statement is confusing since it is also stated that there are several limitations for stem cell organoids.
The title of 3.3.3 “Comparison with 3D bioprinting” may not be adequate since the content of this paragraph is on the benefits of integrating 3D bioprinting.
Line 438, 439, an indentation typo.
Line 449, the paragraph starting with an abbreviation (ECVAM) that has not been mentioned before. This makes the description abrupt. Modification is recommended.
Lines 481 to 486, the meaning is not clear.
In summary, the reviewer recommends the publication of the manuscript after the minor changes.
Author Response
Response to Reviewer 2 Comments
This manuscript reviews the recent progress of organ-on-chip technology, compares its application/performance with conventional 2D and 3D culture on drug development usage, and also discuss the limitation of the technology.
The manuscript is well organized and the description is mostly clear and concise.
The reviewer has some suggestion on minor problems.
The font of the terms in-vivo and in-vitro needs to be consistent. In some places, they are in the plain font but in other places, they are italicized. Whether italicized or not needs to be consistent.
This is an excellent point. We checked 9 in-vivo and 19 in-vitro and revised all to the italicized word to be consistent
In line 27, the meaning of the sentence is not clear: “…data acquired in animal studies is not identical to the mechanisms in humans.”
We intended to express simply in Introduction, and the meaning of the sentence is placed in the last of paragraph of section 2.2 “the use of such inbred mouse models is the lack of heterogeneity” because mouse models are almost homogeneous - (line 75 on page 2 in revised manuscript) However, this suggestion is valid, we deleted the unclear sentence.
In line 66, “…heir therapeutic implications exist, and Malaney et al. reported…”, it is not clear whether a reference is to be inserted.
This is a good point, we decided to move the reference number from the end of the paragraph to the front. A reference is inserted “Malaney et al. [5] reported…”
line 64 on page 2 in revised manuscript
Figure 1 needs some modification in the numbering. There are more than one (c)’s in the figure. This causes confusion in the description in the caption. This issue must be taken care of.
We deleted an extra (c) in Figure 1 on page 7 in revised manuscript
Line 163, it is not clear what do the authors mean by “…fast nature of iPSC…”. A more elaborative description is needed.
We consider that the expression is not essential and deleted the sentence contained an ambiguous expression
line 118 on page 3 in revised manuscript
In lines 274 to 277, it is stated that “… they are less frequently used than stem cell organoids. However, the statement is confusing since it is also stated that there are several limitations for stem cell organoids.
The statement is not the focus of this paragraph so we deleted the sentence.
line 229 on page 5 in revised manuscript
The title of 3.3.3 “Comparison with 3D bioprinting” may not be adequate since the content of this paragraph is on the benefits of integrating 3D bioprinting.
We intended to explain the 3D bioprinting as an advanced technology with organ-on-chip model. The section is composed 3 parts, the first paragraph of the section is introduction about 3D bioprinting as advanced technology, the second is limitation about 3D bioprinting, and the last paragraph is only related to the benefits of integrating 3D bioprining.
Line 438, 439, an indentation typo.
We revised and deleted the wrong paragraph insertion.
line 429 on page 9 in revised manuscript
Line 449, the paragraph starting with an abbreviation (ECVAM) that has not been
mentioned before. This makes the description abrupt. Modification is recommended.
We added “ The European Centre for the Validation of Alternative Methods”
line 439 on page 9 in revised manuscript
Lines 481 to 486, the meaning is not clear.
The paragraph is more ambiguous than intended. We agree that the paragraph contained too much content and references and consider that organ-tumor-on-a-chip is not essential. According to the subtitle of section 4.5 Potential of drug efficacy testing in tumor-on-a-chip and metastasis-on-a-chip, we deleted the paragraph and focused on tumor-on-a-chip and metastasis-on-a-chip.
Instead of the the paragraph, we added recently published references related to efficacy evaluation in OoC (line 479 page on 10 in revised manuscript) and especially focused on Lung and kidney cancer models to the last part of section 4.
In summary, the reviewer recommends the publication of the manuscript after the minor changes.

Reviewer 3 Report
This work provides a review of strengths and limitations of the use of OOC in drug efficacy testing. Unfortunately, I do not feel like recommending your review for publication in Micromachines, even after major revisions.
I find the current work is a weak contribution to the field. As explained in the abstract, the review “attempts to highlight the benefits of OoC as in our understanding of the cellular and molecular pathways in lung and kidney cancer models, and discusses the challenges in evaluating drug efficacy”. I think the text should be extensively improved with more specific examples about lung and kidney cancer applications. For instance, in Chapter 3.2 and subchapters the reader expects specific discussion about liver and kidney cancer.
The overall text should be improved with enhanced citations and discussion about technical details.
The heading 4.2.3 is about biomarkers of lung and kidney cancer without giving information about fluid-based biomarker tests, as the title suggests.
The heading 4.3 and its subheadings are apparently confused and without a logical thread. The titles do not correspond to the content of the subheadings. The text should be better contextualized.
The heading 4.4 contains a very confused paragraph “Ahead… anticancer drugs” (lines 479-494) that should be reframed.
I find the images and tables poorly significant and informative:
- Figure 1 has a lot of experimental data and details that do not contribute to the text but which instead confuse the reader. The caption is not clear and does not explain the aim of the figure in the context of the text. Furthermore, the copyright details and permissions are not declared.
- Table 1 contains speculative concepts and needs to be improved with more technical language. The aspects pointed out in the table should be explained in the text discussing relevant supporting works.
- Table 2 does not contribute to the text. It is a simple list of citations already present along the subheading 4.2.3.
Author Response
Response to Reviewer 3 Comments
This work provides a review of strengths and limitations of the use of OOC in drug efficacy testing. Unfortunately, I do not feel like recommending your review for publication in Micromachines, even after major revisions.
We apologize for this and we have corrected the text as suggested.
I find the current work is a weak contribution to the field. As explained in the abstract, the review “attempts to highlight the benefits of OoC as in our understanding of the cellular and molecular pathways in lung and kidney cancer models, and discusses the challenges in evaluating drug efficacy”. I think the text should be extensively improved with more specific examples about lung and kidney cancer applications.
Several statements that we made were more ambiguous than intended, and we have adjusted the text to be clearer. We have revised and added to improve the text with more specific references.
For instance, in Chapter 3.2 and subchapters the reader expects specific discussion about liver and kidney cancer.
The overall text should be improved with enhanced citations and discussion about technical details.
We intended to describe the OoC: Its basic elements and comparison with other models in section 3 and liver and kidney cancer related references are in section 4
The heading 4.2.3 is about biomarkers of lung and kidney cancer without giving information about fluid-based biomarker tests, as the title suggests.
This is a valid and important question, we intended to clarify of lung and kidney cancer specific biomarker as efficacy evaluation using OoC. However, we are unaware of any studies that provide the answer. So we focused on “Potential” of OoC as a drug efficacy evaluation of lung and kidney cancer model, and we changed the subtitle of section 4 and we also added the word “Potential” to the title.
We would like to describe the potential of efficacy evaluation of lung and kidney cancer model using assays in OoC introduced in earlier.
The heading 4.2 and its subheadings are revised below
4.3. Elements of the drug efficacy evaluation in lung and kidney cancer
4.3.1. Assays for the drug efficacy evaluation in OoC
4.3.2. EGFR and VEGF related molecules as key biomarker of lung and kidney cancer
The heading 4.3 and its subheadings are apparently confused and without a logical thread. The titles do not correspond to the content of the subheadings. The text should be better contextualized.
The heading 4.3 and its subheadings are revised below
4.4 Comparison with conventional efficacy testing methods and institutional devices
4.4.1. Comparison with animal testing
4.4.2. Comparison with clinical trials
4.4.3. Institutional devices as alternatives to animal testing
The heading 4.4 contains a very confused paragraph “Ahead… anticancer drugs” (lines 479-494) that should be reframed.
The paragraph is more ambiguous than intended. We agree that the paragraph contained too much content and references and consider that organ-tumor-on-a-chip is not essential. According to the title of section 4.5 Potential of drug efficacy testing in tumor-on-a-chip and metastasis-on-a-chip, we deleted the paragraph and focused on tumor-on-a-chip and metastasis-on-a-chip.
Instead of the the paragraph, we added recently published references related to efficacy evaluation in OoC (line 479 page on 10 in revised manuscript) and especially focused on lung and kidney cancer models to the last part of section 4.
I find the images and tables poorly significant and informative:
- Figure 1 has a lot of experimental data and details that do not contribute to the text but which instead confuse the reader. The caption is not clear and does not explain the aim of the figure in the context of the text. Furthermore, the copyright details and permissions are not declared. The references of figure and captions are added to legend and figure.
- line 310 on page 7 in revised manuscript
- We added the copyright details and got the permissions after submission.
- Table 1 contains speculative concepts and needs to be improved with more technical language. The aspects pointed out in the table should be explained in the text discussing relevant supporting works.
- We realize that Table 1 is not essential in this paper and specific to efficacy, so we decided to delete, even though Table 1 is from published in nature
- Table 2 does not contribute to the text. It is a simple list of citations already present along the subheading 4.2.3.4.3.2. EGFR and VEGF related molecules as key biomarker of lung and kidney cancer
- line 392 on page 9 in revised manuscript
- This is a good point, we decided to delete table 2 and changed the subtitle of section

Round 2
Reviewer 1 Report
The authors have made significant changes to the manuscript and addressed the majority of my comments. I still feel that the manuscript should include more technical description of existing organ-chips but I understand this may not be the preference of the authors, which is totally fine.
My recommendation is that the Manuscript can be accepted for publication following some minor modifications. Some are detailed below and other are highlighted in the file attached:
Line 20, keywords: Organ-on-chip is listed twice. And please don't include "organs-on-a-chips", it just doesn't make sense. "Organ-chips", "organs-on-chips", "organ-on-a-chip" are all better options in my opinion.
Line 25: "a lot of" is quite informal. "many" would be better
lines 45/46: "2D cell culture methods are generally utilised to screen toxic compounds". This just repeats the information from the previous sentence. I would remove it.
line 127: "Until now, this issue was addressed culture media[18]." This sentence doesn't make sense.
line 144: "eukaryotes" should be changed to "eukaryocytic cells". People are eukaryotes and we are not surrounded by ECM, our cells are.
lines 224, 225: "Because of the limitations in the expandability, availability, and handling capacity of tissues needed for somatic-cell organoids" Sentence is disconnected and doesn't make sense. Sentences shouldn't start with "Because".
lines 269, 270: "OoC engineering will facilitate the modeling of tissue-specific functions, micro-organs with heterogeneity, a desired 3D cellular arrangement". Please re-write.
Line 282: "This in vitro model, OoC model". Model twice?
line 292: There should be a full stop after 2018.
Figure 1 caption is too long, perhaps simplifying the 'Source' description would help? It's not common to see a full reference citation in Figure captions.
line 338: first in-human trials
line 371: "These trials only indicated..." "were stained" "could not". I still don't understand the use of the past tense here, because the text is not referring to a particular occurrence but to the techniques in general, correct?
Line 373: Wrong use of the conjunction "While". Please re-phrase.
line 394: "and stimulating" ... what?
line 462: "to improvement". Improvement is not a verb.
Lines 481, 482: "reported..." "was developed". unnecessary verb repetition.

Author Response
The authors have made significant changes to the manuscript and addressed the majority of my comments. I still feel that the manuscript should include more technical description of existing organ-chips but I understand this may not be the preference of the authors, which is totally fine.
We are very grateful for this precious comment and appreciate the positive responses.
My recommendation is that the Manuscript can be accepted for publication following some minor modifications. Some are detailed below and other are highlighted in the file attached:
We decided to delete the highlighted sentence of the line 260 on page 6
“There are not many tested bioink compositions currently.”
We realized this sentence is not essential.
Line 20, keywords: Organ-on-chip is listed twice. And please don't include "organs-on-a-chips", it just doesn't make sense. "Organ-chips", "organs-on-chips", "organ-on-a-chip" are all better options in my opinion.
We deleted the first word "organs-on-a-chips" in keywords and selected the same as the title “organ-on-a-chip” as you give us your opinion
Line 25: "a lot of" is quite informal. "many" would be better
We changed the better word 'many'.
lines 45/46: "2D cell culture methods are generally utilised to screen toxic compounds". This just repeats the information from the previous sentence. I would remove it.
This is a good point, we decided to remove the sentence.
line 127: "Until now, this issue was addressed culture media[18]." This sentence doesn't make sense.
This is an excellent point, we decided to delete the sentence.
line 144: "eukaryotes" should be changed to "eukaryocytic cells". People are eukaryotes and we are not surrounded by ECM, our cells are.
We apologize for this and I corrected the other file, maybe.
We changed the better word "eukaryocytic cells ' line 142 on page 3 in revised manuscript
lines 224, 225: "Because of the limitations in the expandability, availability, and handling capacity of tissues needed for somatic-cell organoids" Sentence is disconnected and doesn't make sense. Sentences shouldn't start with "Because".
We consider that the sentence is not essential and should be deleted.
line 223 on page 5 in revised manuscript
lines 269, 270: "OoC engineering will facilitate the modeling of tissue-specific functions, micro-organs with heterogeneity, a desired 3D cellular arrangement". Please re-write.
We agreed and removed the ambiguous expression “a desired 3D cellular arrangement”
Line 282: "This in vitro model, OoC model". Model twice?
Unintentionally, “model” was used two times.
We revised to “This in vitro model, OoC represents”
line 278 in revised manuscript.
line 292: There should be a full stop after 2018.
We revised that “Yang et al. [54] developed a Lung-on-a-chip in 2018. The chip is simple, effective, and easy to utilize”
line 288 in revised manuscript.
Figure 1 caption is too long, perhaps simplifying the 'Source' description would help? It's not common to see a full reference citation in Figure captions.
This is an important suggestion, we simplified the 'Source' description below
Journal abbreviated title, publication year, [reference number]
Line 314-324, on page 7 in revised manuscript.
line 338: first in-human trials
we removed the other hyphen.
Line 330 on page 7 in revised manuscript.
line 371: "These trials only indicated..." "were stained" "could not". I still don't understand the use of the past tense here, because the text is not referring to a particular occurrence but to the techniques in general, correct?
This is a valid comment, we intended to express the techniques in general and revised the sentences below.
These trials only indicate whether the cells survived, but they are unable to monitor the cells during real-time long-term recordings. In addition, once the cells are stained, they are not be used for further treatments.
Line 364 on page 8 in revised manuscript.
Line 373: Wrong use of the conjunction "While". Please re-phrase.
The conjunction is more ambiguous than intended, so we deleted “while” and added “The” in front of the sentence.
Line 365 on page 8 in revised manuscript.
line 394: "and stimulating" ... what?
We realized the word “stimulating" is not essential in the sentence, so we decided to remove "and stimulating"
line 462: "to improvement". Improvement is not a verb.
This is a good point, we decided to remove “ment”
Lines 481, 482: "reported..." "was developed". unnecessary verb repetition.
This is a good point, we decided to delete the unnecessary verb "was developed".

Reviewer 3 Report
The authors have made considerable improvements in content and clarity of the manuscript in response to the review. Overall, all the suggestions and corrections have been well reflected throughout the new manuscript, but some minor things should be corrected before accepting the manuscript.
- OOC and MPS are used interchangeably but could be ambiguous. Please use consistent terminology
- Chapters 2 and 3 (line 40-278) still have few references that could help the reader to investigate further. In particular, the text should be improved with citations of significant articles that show the experimental evidence of the concepts explained.
Author Response
The authors have made considerable improvements in content and clarity of the manuscript in response to the review. Overall, all the suggestions and corrections have been well reflected throughout the new manuscript, but some minor things should be corrected before accepting the manuscript.
We are very grateful for this precious comment and appreciate the positive responses.
- OOC and MPS are used interchangeably but could be ambiguous. Please use consistent terminology
We used the consistent terminology “OoC” instead of 9 MPS in the manuscript to remove the ambiguous part.
- Chapters 2 and 3 (line 40-278) still have few references that could help the reader to investigate further. In particular, the text should be improved with citations of significant articles that show the experimental evidence of the concepts explained.
We apologize for this, I tried to add the references in the previous manuscript
and I was requested that we need more specific references in Chapters 4
because there are already more references in Chapters 2 and 3 than Chapters 4 in the manuscript.
We intended to express comprehensive contents about OoC in Chapters 2 and 3 and to focus on Chapters 4 which is directly related to the title.
